# Dual-Signal-Encoded Barcodes with Low Background Signal for High-Sensitivity Analysis of Multiple Tumor Markers

**Bo Zhang, Wan-Sheng Tang and Shou-Nian Ding \***

School of Chemistry and Chemical Engineering, Southeast University, Nanjing 211189, China;
zhangbo@seu.edu.cn (B.Z.); wstang@seu.edu.cn (W.-S.T.)

\* Correspondence: snding@seu.edu.cn; Fax: +86-25-5209-0621

**Abstract:** The suspension array technology (SAT) is promising for high-sensitivity multiplexed analysis of tumor markers. Barcodes as the core elements of SAT, can generate encoding fluorescence signals (EFS) and detection fluorescence signals (DFS) in the corresponding flow cytometer channel. However, the bleed-through effect of EFS in the DFS channel and the reagent-driven non-specific binding (NSB) lead to background interference for ultrasensitive assay of multiple targets. Here, we report an ingenious method to eliminate background interference between barcode and reporter using low-background dual-signal-encoded barcodes (DSBs) based on microbeads (MBs) and quantum dots (QDs). The low-background DSBs were prepared via combination strategy of two signals containing scatter signals and fluorescence signals. Three types of MBs were distinguished by the scattering channel of flow cytometer (FSC vs. SSC) to obtain the scattered signals. Green quantum dots (GQDs) or red quantum dots (RQDs) were coupled to the surface of MBs by sandwich immune structure to obtain the distinguishable fluorescent signals. Furthermore, the amount of conjugated capture antibody on the MB's surface was optimized by comparing the change of detection sensitivity with the addition of capture antibody. The combination measurements of specificity and NSB in SAT platform were performed by incubating the capture antibody-conjugated MBs (cAb-MBs) with individual QD-conjugated detection antibody (QDs-dAb). Finally, an SAT platform based on DSBs was successfully established for highly sensitive multiplexed analysis of six tumor markers in one test, which suggests the promising tool for highly sensitive multiplexed bioassay applications.

**Keywords:** dual-signal-encoded barcodes; background interference; microbeads; flow cytometer; six tumor markers

## 1. Introduction

Cancer, especially terminal cancer with high mortality rate, is considered one of the deadliest diseases in the world [1]. Diagnosing and treating cancer at an early stage can effectively inhibit the deterioration of the disease and avoid the appearance of terminal cancer [2–5]. However, a serious obstacle to diagnosing cancer at an early stage is lack of the incipient symptoms [4]. Fortunately, some small biomolecules from blood or tissue can be used as tumor markers for early diagnosis of cancer, notably their abnormally high concentrations indicate that certain cancers are likely to be present in the human body [6,7]. Moreover, in order to efficiently and accurately diagnose cancer, multiple tumor markers are often required to be detected simultaneously because it is difficult to diagnose cancer by using a single tumor marker with insufficient specificity [4]. Hence, simultaneous detection of multiple tumor markers plays an important role in early screening of cancer, assessment of disease severity, and accurate analysis of cancer treatment effects [4,6,8,9].

Until now, various analytical approaches have been applied to simultaneous detection of multiple tumor markers, for instance, surface plasmon resonance (SPR) [10], photoluminescent [11], surface-enhanced Raman scattering (SERS) [12], electrochemistry [13–15], and fluorescence [4,16–18]. In particular, SAT based on fluorescence-encoded microbeads

with fast binding kinetics, low sample volume, high detection sensitivity, and excellent multiplexing capability is one of the most promising fluorescence analysis methods in the field of the high-sensitivity simultaneous analysis of multiple targets [19–21]. SAT enables high-throughput simultaneous detection of dozens of analytes on flow cytometers, such as simultaneous detection of up to 35 analytes containing tumor markers and cytokines in small sample volumes [22]. Although SAT has many advantages mentioned above, its performance still needs to be further improved and the following three challenges need to be addressed [23]: (1) limited number of barcodes, (2) unpredictable barcode signal, and (3) suboptimal detection sensitivity. Most of the published work on SAT performance improvement focused on challenges (1) and (2). For example, the orthogonal fluorescence signals were obtained by inhibited energy transfer based on CdSe/CdS quantum dots without spectral overlap and 144 coded microbeads-based barcodes were successfully prepared by the Shirasu porous glass membrane emulsification method [16]. The non-orthogonal fluorescence signals were effectively predicted by ensemble multicolor Förster resonance energy transfer (emFRET) model to guide barcoding [22]. The barcode library with a total of 580 barcodes was established using four dyes [22]. A total of 300 superparamagnetism barcodes were established by structure-fluorescence combinational encoding strategy [18].

However, SAT platform is difficult to detect ultralow concentration samples due to background interference and suboptimal detection sensitivity. Especially, for SAT based on fluorescence-encoded microbeads, the bleed-through effect of EFS in the DFS channel of flow cytometry and the reagent-driven non-specific binding (NSB) lead to background interference [24–26]. Although the bleed-through effect of EFS in the DFS channel of flow cytometry is common, its impact on detection sensitivity is neglected. Moreover, bovine serum albumin (BSA) is widely used to inhibit nonspecific binding, but combination measurements of specificity and NSB (including cross-reactivity) in SAT platform are rarely performed [25,26]. To the best of our knowledge, no report has demonstrated the reduction of background interference to improving detection sensitivity by preventing the bleed-through effect of EFS in the DFS channel of flow cytometry.

Herein, we report an ingenious method to eliminate the bleed-through effect of EFS in the DFS channel of flow cytometry by using low-background DSBs. As illustrated in Scheme 1, low-background DSBs were prepared via combination strategy of scatter signals and fluorescence signals. The scatter signals come from the forward and side scatter channels of the flow cytometer. The fluorescence signals come from FL1-H and FL2-H channels of the flow cytometer. The low-background DSBs are the basis for simultaneous identification and high-sensitivity quantitative analysis of multiple targets. Three types of MBs, including $SiO_2$-L, PS-L, and PS-S (L: large particle size (>5 μm), S: small particle size (<3 μm)), were distinguished by the scattering channel of flow cytometer (FSC vs. SSC) to obtain the scattered signals, as shown in Scheme 1. Green quantum dots (GQDs) or red quantum dots (RQDs) were coupled to the surface of MBs by sandwich immune structure to obtain the distinguishable fluorescent signals, as shown in Scheme 2a,b. Furthermore, the amount of conjugated capture antibody on the MBs surface was optimized by comparing the change of detection sensitivity with the addition of capture antibody. The combination measurements of specificity and NSB in SAT platform were performed by incubating the cAb-MBs with individual QDs-dAbs. Finally, the high-sensitivity analysis abilities of the DSBs to tumor markers were demonstrated via multiplex sandwich immunoassays. SAT platform were established for simultaneous multiplexed detection of CEA, CA125, SCCA, AFP, NSE, and CA724. These developed DSBs, with low background signal for high-sensitivity analysis of multiple tumor markers, would offer a new platform for early, efficient, and accurate diagnosis of cancer.

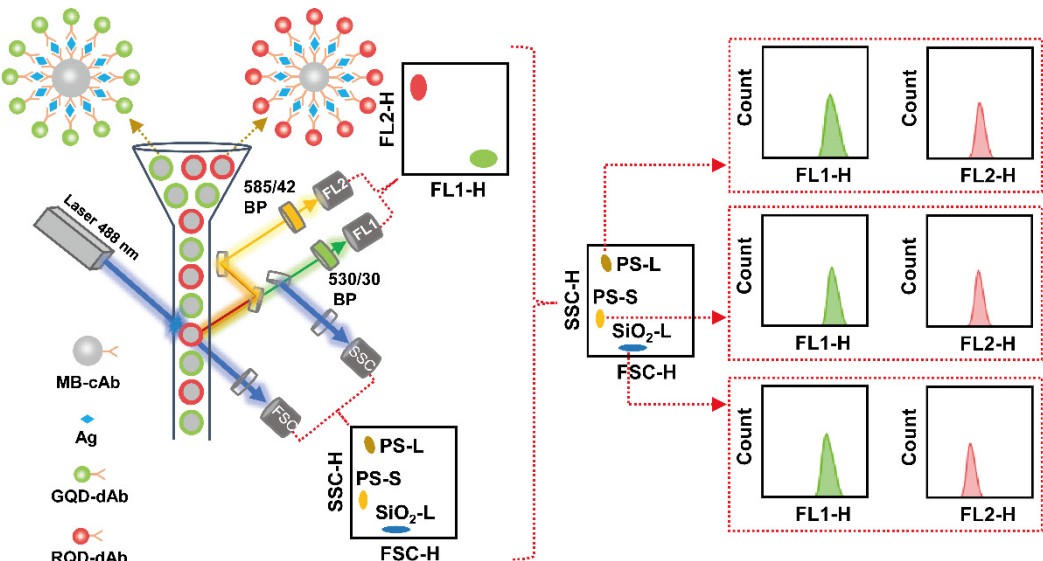

**Scheme 1.** Schematic illustration of the flow cytometric readout microbeads and the dual-signal-encoded strategy.

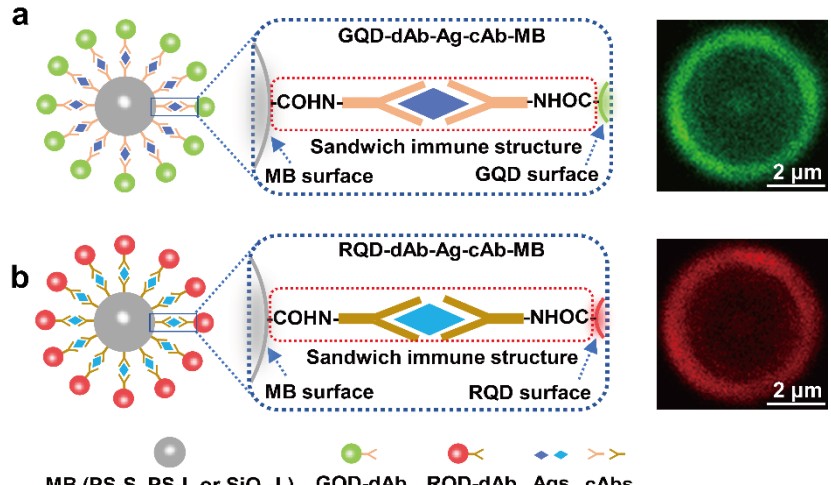

**Scheme 2.** Construction of DSBs. (**a**) GQDs or (**b**) RQDs were coupled to the surface of MBs by sandwich immune structure. Laser-scanning confocal images of corresponding DSBs are on the right.

## 2. Experimental Section

### 2.1. Reagents and Chemicals

Carcinoembryonic antigen (Ag-CEA), carcinoembryonic antibody (cAb-CEA, dAb-CEA), carbohydrate antigen 125 (Ag-CA125), carbohydrate antibody 125 (cAb-CA125, dAb-CA125), squamous cell carcinoma antigen (Ag-SCCA), squamous cell carcinoma antibody (cAb-SCCA, dAb-SCCA), alpha fetoprotein antigen (Ag-AFP), alpha fetoprotein antibody (cAb-AFP, dAb-AFP), neuron specific enolase antigen (Ag-NSE), neuron specific enolase antibody (cAb-NSE, dAb-NSE), carbohydrate antigen 724 (Ag-CA724), and carbohydrate antibody 724 (cAb-CA724, dAb-CA724) were purchased from Linc-Bio Science Co., Ltd. (Shanghai, China). All other reagents were shown in supplementary chemical and reagents.

### 2.2. Characterization

Scanning electron microscope (SEM) images of MBs were obtained with a Schottky field emission scanning electron microscope (FEI, Inspect F50, Hillsboro, OR, USA) at 10 kV. Photoluminescence spectra were measured by spectrofluorometer (Shimadzu, RF-6000,

Kyoto, Japan). Fluorescence images of MBs were photographed by a super-resolution multiphoton confocal microscope (LSM 700, Leica Microsystems, Wetzlar, Germany) at an excitation wavelength of 405 nm. Barcode signals were decoded using a flow cytometer (FACSCalibur, Becton Dickinson, Franklin Lakes, NJ, USA).

### 2.3. Synthesis of CdTe QDs

Two kinds of CdTe QDs, green QDs (GQDs) and red QDs (RQDs), were prepared as described in the literature [27]. Briefly, 40 mL, pH = 12, mixed solution, containing 67.0 μL of MPA and 91.3 mg of $CdCl_2 \cdot 2.5H_2O$, and 1.0 mL 0.04 M of NaHTe solution were prepared in two three-necked flasks under stirring with $N_2$, respectively. NaHTe solution was added to the mixed solution and then refluxed at 100 °C. The CdTe QDs with emission wavelengths of 522 nm (GQDs) and 630 nm (RQDs) were obtained by changing the reflow time, respectively, and then extracted from the crude solution by centrifugation at 10,000 rpm for 10 min, purified three times with ethanol and $H_2O$, and finally dispersed in deionized water.

### 2.4. Synthesis of Polystyrene MBs

First, 1.496 g poly(vinylpyrrolidone) (PVP) was dissolved in 50 mL mixed solution (27 mL ethanol and 18 mL 2-methoxyethanol) in a 250 mL three-necked flask with a magnetic stirrer and then 13.804 g styrene solution (13.6 g styrene and 0.204 g 2,2-azobis(isobutyronitrile) (AIBN)) was added to the mixed solution. The mixed solution was degassed with a stream of $N_2$ for 10 min, then heated to 45 °C for 30 min, finally kept at 70 °C for 12 h in order to obtain PS-S. The white solid power, PS-Ss, were subsequently obtained from the mixed solution after the reaction by centrifugation, washing with ethanol, and drying overnight in a vacuum oven.

### 2.5. A Silica Shell Coating and Functionalization of Microbeads

First, 0.020 g of PSs were ultrasonically dispersed in the mixture containing 20 mL of ethanol, 0.1 mL of $H_2O$, and 10 μL of TEOS. After being added to 5 μL of $NH_3 \cdot H_2O$, the mixture was further reacted at 25 °C for 24 h. Then the products (PS-SiO$_2$) were separated from the mixture by centrifugation and respectively washed three times with water and ethanol [28].

Afterwards, functionalization of the PS-SiO$_2$, that is, amination and carboxylation, was carried out as follows [29]. The MBs were dispersed in 5 mL of ethanol, and then 100 μL of APTES dilution was added under stirring at 40 °C for 4 h. The products of amino-functionalized MBs (PS-SiO$_2$-NH$_2$) were washed with ethanol and MEST (10 mM, pH 5.0) three times, respectively. Next, the MBs (PS-SiO$_2$-NH$_2$) were redispersed in 5 mL of MEST, and then by the addition of 62.5 mg of PAA and 10.0 mg of EDC. After stirring the reaction for 2 h, the carboxylated MBs (PS-SiO$_2$-COOH) were centrifuged and washed three times with water. Herein, PS MBs (PS-S, PS-L) and SiO$_2$ MBs (SiO$_2$-L) were functionalized and finally named PS-S-COOH, PS-L-COOH, and SiO$_2$-L-COOH, respectively.

### 2.6. Bioconjugation of QDs with dAb

The bioconjugation method of QDs (GQDs or RQDs) with dAb was similar to previous literature [30,31]. Taking the bioconjugation procedure of GQDs with dAb-CEA as an instance, 75 μL of EDC solution (4.2 mg/mL) was added to 150 μL of GQDs (3 mg/mL) with shaking for 0.5 h at room temperature in dark, and then 10 μL of dAb-CEA (2.6 mg/mL) was added to the above mixture at room temperature in the dark. Finally, the conjugated product, GQDs-dAb-CEA as a probe for quantitative detection, was purified by ultrafiltration with 50 KD filter to remove unconjugated GQDs, and then stored at 4 °C for further use.

### 2.7. Bioconjugation of MBs with cAb

For six-plexed detection of tumor markers, six coupling structures of cAbs-MB (cAbs-CEA-SiO$_2$-L, cAbs-CA125-SiO$_2$-L, cAbs-SCCA-PS-L, cAbs-AFP-PS-L, cAbs-NSE-PS-S, and cAbs-CA724-PS-S) were prepared by bioconjugation of the carboxylated MB with cAb. Briefly, the functionalized MBs were activated in activation buffer containing NHS and EDC, then they were washed by using PBS (pH = 7.4) with centrifugation. Next, $2 \times 10^4$ of activated MBs and a certain amount of cAb were incubated in 500 µL PBS at 10 °C for 12 h. Afterwards, MBs were washed with PBS to remove unreacted Abs and then blocked with BSA. Finally, the obtained cAbs-MB was stored in 0.1% BSA at 4 °C for later use.

### 2.8. 6-Plex Sandwich Immunoassay of Tumor Biomarkers

The six-plex sandwich immunoassay of tumor biomarkers (CEA, CA125, SCCA, AFP, NSE, and CA724) was performed in a 96-well plate. First, 100 µL of the prepared antigen (Ag) solution at different concentrations (0, 0.001, 0.01, 0.1,1, 10, 100, and 1000 KU/L "for CA125 and CA724" as well as ng/mL "for CEA, SCCA, AFP and NSE") were added to each well of the 96-well plate.

Then, $2 \times 10^4$ of cAbs-MBs were mixed with the Ag solution and the mixture was further incubated at room temperature for 1 h in the dark, then washed three times with wash buffer. Subsequently, 100 µL of QDs-dAb was added to each well, and the mixture was incubated at room temperature for 1 h, and then the excess QDs-dAb was removed to obtain QDs-dAb-Ag-cAbs-MB (GQD-CEA-SiO$_2$-L, RQD-CA125-SiO$_2$-L, GQD-SCCA-PS-L, RQD-AFP-PS-L, GQD-NSE-PS-S, RQD-CA724-PS-S). Finally, the multiplexed detection results were analyzed by flow cytometry.

## 3. Results and Discussion

### 3.1. Construction of DSBs with Low Background

Background interference of barcodes, for instance, the non-specific binding effects of reagents on the microbead surface and the bleed-through effect of encoding fluorescence signals in the detection fluorescence signals channel of flow cytometry, is one major issue facing SAT in high-sensitivity multiplexed analysis of tumor markers. In this research, dual-signal barcodes with low background were used to suppress the bleed-through effect of encoding fluorescence signals to achieve higher detection sensitivity for SAT. Low-background DSBs based on MBs and QDs were prepared via a combination strategy of two signals containing scatter signals and fluorescence signals. The scatter signals come from the forward and side scatter channels of the flow cytometer (Scheme 1 and Figure S3 in Supplementary Materials). The fluorescence signals come from FL1-H and FL2-H channels of the flow cytometer (Scheme 1 and Figure S3 in Supplementary Materials). The low-background DSBs are the basis for simultaneous identification and high-sensitivity quantitative analysis of multiple targets.

First, three types of MBs (SiO$_2$-L, PS-L, and PS-S), different particle sizes and materials (Figures S1 and S2), were distinguished by the scattering channel of flow cytometer (FSC vs. SSC) to obtain the scattered signals that can effectively reduce barcode background and ensure high detection sensitivity because they are non-fluorescent signals that do not interfere with fluorescent detection signals, as shown in Scheme 1. Then, GQDs or RQDs were coupled to the surface of MBs by sandwich immune structure to obtain the distinguishable fluorescent signals in the FL1-H or FL2-H channels of flow cytometer, as shown in Schemes 1 and 2 and Figure S3 in Supplementary Materials. It is worth noting that the distinguishable fluorescent signals have the dual function of identifying and quantifying the analyte targets, and DSBs were also read out using a flow cytometer with a blue 488 nm laser, which is simple and efficient for the decoding process.

The ability to distinguish between barcodes is a necessary condition for multiple analysis in SAT. In order to verify the feasibility of the dual-signal coding strategy, as a proof of concept, six kinds of DSBs (GQDs-CEA-SiO$_2$-L, RQDs-CA125-SiO$_2$-L, GQDs-SCCA-PS-L, RQDs-AFP-PS-L, GQDs-NSE-PS-S, RQDs-CA724-PS-S) with two-signal combinations are

measured by flow cytometer, as shown in Figures 1 and S4 in Supplementary Materials. First, the scatter signal barcodes of SiO$_2$-L, PS-L, and PS-S can be clearly distinguished by the scatter plot of SSC versus FSC in the flow cytometer, as shown in Figure 1a. Next, six kinds of fluorescence signals barcodes can be clearly distinguished by histograms corresponding of FL1-H and FL2-H channels, as shown in Figure 1b–d. Therefore, the DSBs constructed with the dual-signal coding strategy can be used for multiplex analysis of SAT.

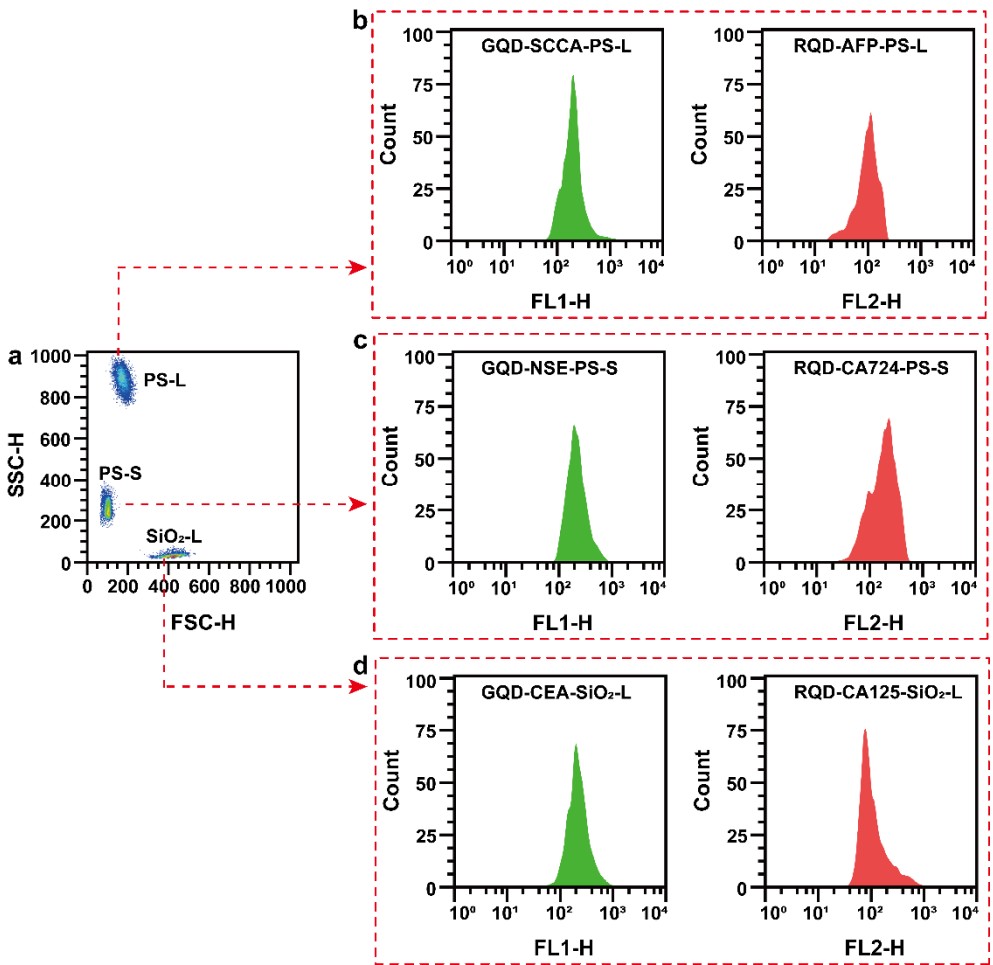

**Figure 1.** DSBs distinguishability validation. (**a**) The scatter signal barcodes of SiO$_2$-L, PS-L, and PS-S can be distinguished by the scatter plot of SSC versus FSC. Histograms of three groups of fluorescent signal barcodes. (**b**) GQD-SCCA-PS-L and RQD-AFP-PS-L, (**c**) GQD-NSE-PS-S and RQD-CA724-PS-S, and (**d**) GQD-CEA-SiO$_2$-L and RQD-CA125-SiO$_2$-L.

### 3.2. Optimization of the Addition Amount of cAb

The research of early diagnosis, treatment, and prognosis of cancer is one of the most active areas of biological analysis, and SAT, due to its high multiplexing capacity and high-throughput analysis of low sample volume, is one of the popular choices for the highly sensitive detection of important tumor markers [32–34]. Ag-CEA, Ag-CA125, Ag-SCCA, Ag-AFP, Ag-NSE, and Ag-CA724 play important roles in identifying early malignant tumors including lung cancer, ovarian cancer, cervical cancer, liver cancer, gastrointestinal cancer, among others [33,35]. Hence, a highly sensitive Ag-CEA, Ag-CA125, Ag-SCCA, Ag-AFP, Ag-NSE, and Ag-CA724 assay is of great importance for early diagnosis, treatment, and prognosis of cancer in practice clinical samples. For effective establishment of the highly sensitive suspension assay, the amount of conjugated capture antibody on the MB's surface was optimized by comparing the change of detection sensitivity with the addition of capture antibody.

A series of different additions of cAb-CEA (25, 250, 500, 1000, 2500, and 5000 ng) under $2 \times 10^4$ MBs and 0.5 µg GQDs-dAb-CEA conditions were sequentially carried out in order to obtain the optimum addition amount of cAb-CEA. Histograms of GQD-CEA-SiO$_2$-L barcodes were measured by FL1-H detection channel of the flow cytometry, as shown in Figure 2a–f. It can be seen in Figure 2a–f that the GQD-CEA-SiO$_2$-L barcode position is obviously shifted to the right with increasing amounts of cAb-CEA in the range 25−500 ng, indicating that the addition of cAb-CEA in the range 25−500 ng has a significant effect on the MFI values of GQD-CEA-SiO$_2$-L barcode and the detection sensitivity of barcodes for tumor markers. The MFI values of GQD-CEA-SiO$_2$-L barcodes increased significantly with the addition amount of cAb-CEA in the range 25−500 ng, and the MFI values increased less in the range of 500−5000 ng. Therefore, when the addition amount of cAb-CEA is 500 ng, enough cAb-CEA is bound on the surface of MBs, as shown in Figure 2g.

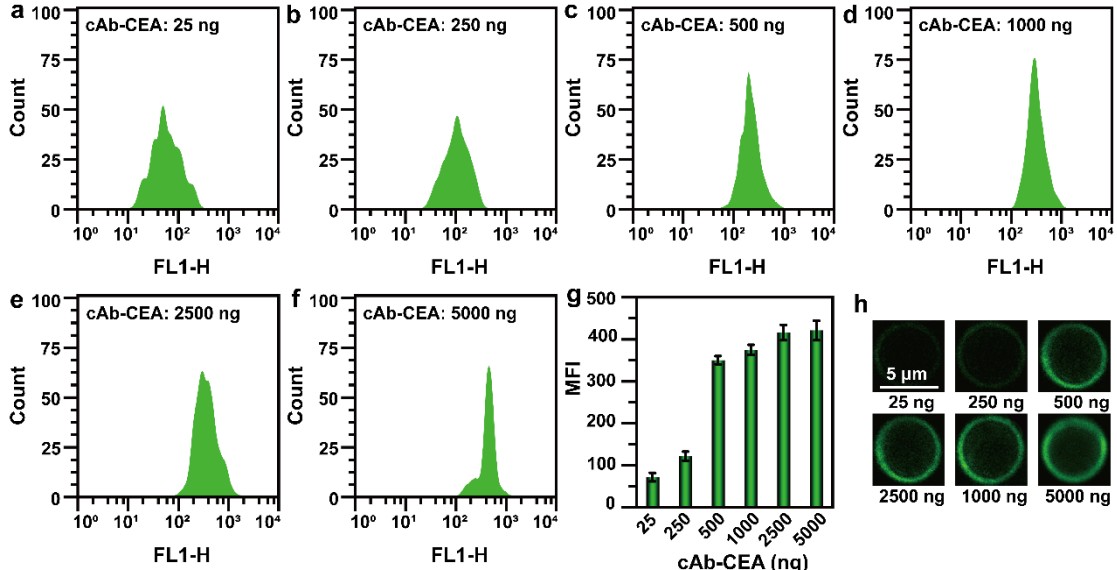

**Figure 2.** Optimization of added amounts of cAb-CEA. Histograms of GQD-CEA-SiO$_2$-L barcodes with (**a–f**) 25, 250, 500, 1000, 2500, and 5000 ng of cAb-CEA, respectively. (**g**) Optimal amount of antigen of cAb-CEA (500 ng). (**h**) Laser-scanning confocal images of GQD -CEA-SiO$_2$-L with different amounts of cAb-CEA.

To further confirmation, the results of the laser scanning confocal microscopy (LSCM) images of GQD-CEA-SiO$_2$-L after adding 25–5000 ng of cAb-CEA are presented in Figure 2h, indicating that the fluorescence intensity of GQD-CEA-SiO$_2$-L increased with the addition amount of cAb-CEA, which is consistent with the findings in Figure 2a–g. Therefore, in order to establish the highly sensitive suspension assay based on cAb-saving, 500 ng of cAb-CEA was taken as the optimum addition amount.

In order to optimize the binding amount of cAb-CA125 on the surface of SiO$_2$-L, a series of different additions of cAb-CA125 (36, 360, 720, 1440, 3600, and 7200 ng) under $2 \times 10^4$ MBs and 0.5 µg RQDs-dAb-CA125 conditions were sequentially carried out under the same experimental conditions as cAb-CA125. Histograms of RQD-CA125-SiO$_2$-L barcodes were measured by FL2-H detection channel of the flow cytometry, as shown in Figure 3a–f. It can be seen in Figure 3a–f that the RQD-CA125-SiO$_2$-L barcode position is obviously shifted to the right with increasing amounts of cAb-CA125 in the range 36−1440 ng, indicating that the addition of cAb-CA125 in the range 36−1440 ng has a significant effect on the MFI values of RQD-CA125-SiO$_2$-L barcodes and the detection sensitivity of barcodes for tumor markers. The MFI values of RQD-CA125-SiO$_2$-L barcode increased significantly with the addition amount of cAb-CA125 in the range 36–1440 ng and retained a high value in the range of 1440−7200 ng, which indicates that enough

cAb-CA125 is bound on the surface of MBs when the addition amount of cAb-CEA is 1440 ng, as shown in Figure 3g.

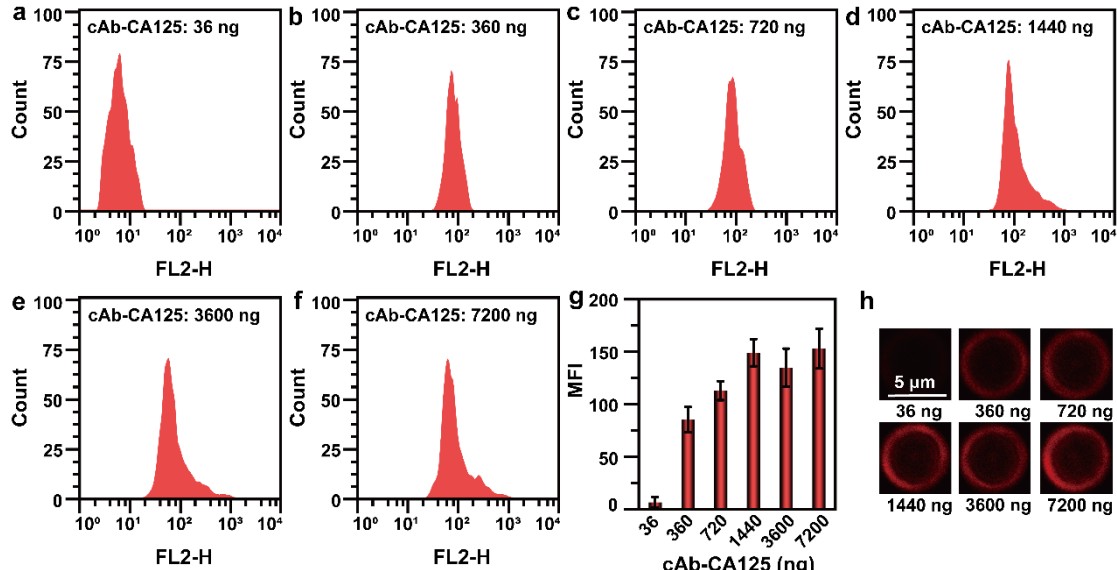

**Figure 3.** Optimization of added amounts of cAb-CA125. Histograms of RQD-CA125-SiO$_2$-L barcodes with (**a–f**) 36, 360, 720, 1440, 3600, and 7200 ng of cAb-CA125, respectively. (**g**) Optimal amount of antigen of cAb-CA125 (1440 ng). (**h**) Laser-scanning confocal images of GQD-CA125-SiO$_2$-L with different amounts of cAb-CA125.

Furthermore, the results of the LSCM images of RQD-CA125-SiO$_2$-L after adding 36−7200 ng of cAb-CA125 are presented in Figure 3h, which is consistent with the findings in Figure 3a–g. Hence, 1440 ng of cAb-CA125 was taken as the optimum addition amount. It is worth noting that the fluorescence intensity of RQD-CA125-SiO$_2$-L is smaller than that of GQD-CEA-SiO$_2$-L after optimizing the cAb, mainly because the emission peak of GQD has a better position in the corresponding detection channel of the flow cytometry (Figure S3 in Supplementary Materials). Histograms of the MFI values of GQD-SCCA-PS-L, RQD-AFP-PS-L, GQD-NSE-PS-S, and RQD-CA724-PS-S barcodes are shown in Figure S5a–d in Supplementary Materials, which indicates that the optimum addition amounts of cAb-SCCA, cAb-AFP, cAb-NSE, and cAb-CA724 are 1320, 640, 500, and 1000 ng, respectively.

### 3.3. Non-Specific Binding

In multiplex sandwich immunoassays, there is mainly specific binding between cAb-MBs, Ag (tumor marker), and QDs-dAb. However, reagent-driven non-specific binding (NSB) is prone to occur during incubation of cAb-MBs, Ag, and QDs-dAb, which is a major problem facing SAT in highly sensitive multiplexed analysis of tumor markers, decreasing detection sensitivity. Therefore, the specific binding between cAbs-MBs, Ags, and QDs-dAbs requires individual and combined validation to ensure high detection sensitivity. The specific binding and NSB in a 6-plex immunoassay with six combinatorial interactions (CEA, CA125, SCCA, AFP, NSE, and CA724) were conducted using MBs (SiO$_2$-L, PS-L, and PS-S). To uncover NSB events, pools of cAb-MBs against the six tumor makers (Ags) were incubated in different wells with different assay reagents, including cAb-MBs, Ags, and QDs-dAbs, as shown in Figure S6a–c (Supplementary Materials). The cross-linking reaction between cAbs-MBs and QDs-dAbs was detected by incubating the cAb-MBs with individual QDs-dAbs, and then identified by flow cytometry. As show in Figure S7 (Supplementary Materials), there is no obvious cross-linking reaction signal, which indicates that weaker NSB between cAbs-MBs and QDs-dAbs.

To study Ag binding, a series of combinatorial experiments were conducted by incubating the cAbs-MBs with individual Ag solution at 1 ng/mL (or KU/L) or 100 ng/ mL (or KU/L), followed by QDs-dAbs. As show in Figure 4a,b, specific binding signals of cAbs-MBs and Ag were strong but NSB signals were weak, which indicates that NSB signals of cAbs-MBs and Ag are not concentration dependent. These results demonstrate that NSB events can be efficiently recognized by a series of combinatorial experiments. Especially, compared with strong specific detection signals, the weak signals of detection background can effectively ensure highly sensitive multiplexed analysis of tumor markers.

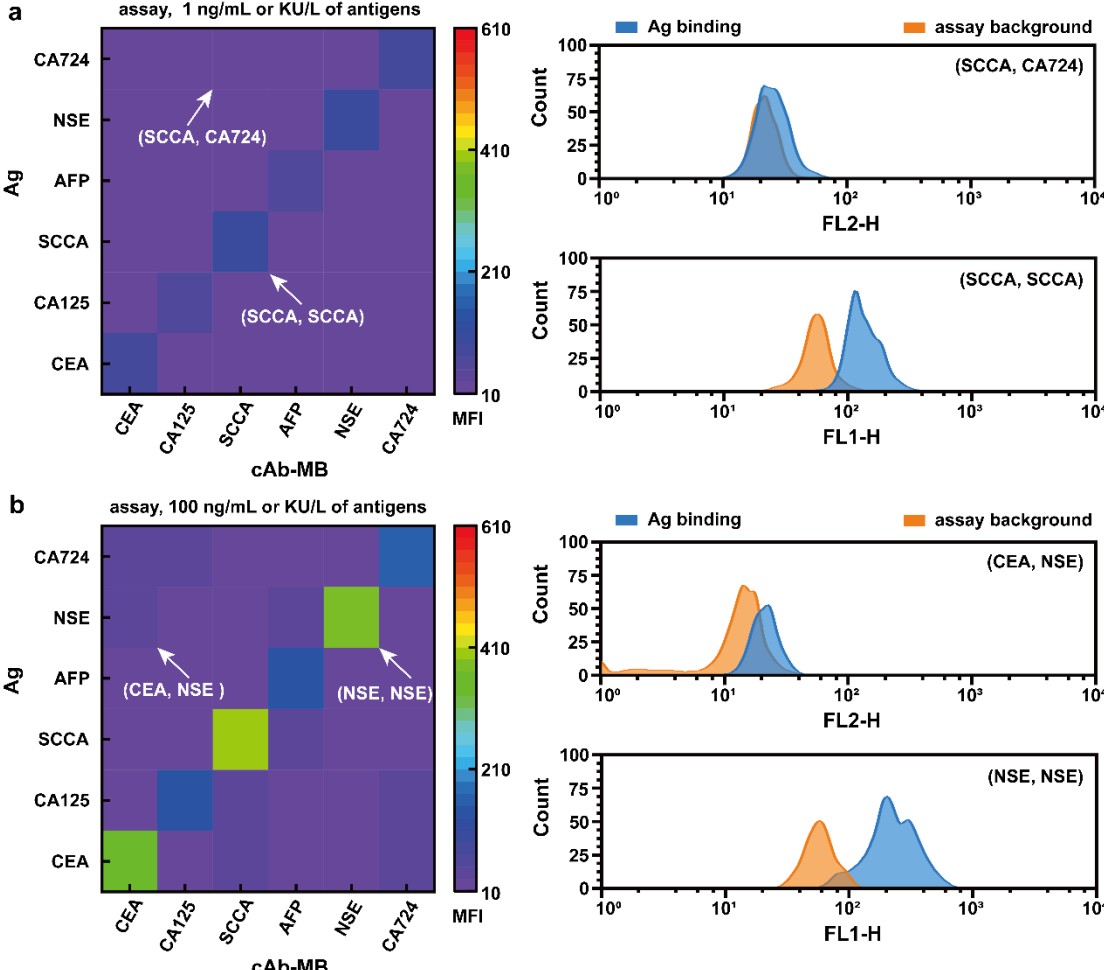

**Figure 4.** Combinatorial experiments of 6-plex specific and non-specific binding. (**a,b**) cAb-MB (column) and Ag (row) incubation and then addition of QDs-dAb. Diagonal and off-diagonal represent specific and non-specific antigen binding, respectively. Histograms of the count for (cAb, Ag) are on the right.

### 3.4. Application of DSBs in High-Sensitivity Analysis of Multiple Tumor Markers

To demonstrate the practical capability of DSBs for highly sensitive multiplexed analysis of tumor markers, Ag-CEA, Ag-CA125, Ag-SCCA, Ag-AFP, Ag-NSE, and Ag-CA724, as a proof-of-concept, barcodes with scattering signals coded addresses of $SiO_2$-L, PS-L, and PS-S were conjugated with specific capture antibodies as follow, cAbs-CEA-$SiO_2$-L, cAbs-CA125-$SiO_2$-L, cAbs-SCCA-PS-L, cAbs-AFP-PS-L, cAbs-NSE-PS-S, and cAbs-CA724-PS-S. Furthermore, barcodes with fluorescence signals coding addresses of GQDs and RQDs were conjugated with detection antibodies as follow, GQDs-dAb-CEA, RQDs-dAb-CA125, GQDs-dAb-SCCA, RQDs-dAb-AFP, GQDs-dAb-NSE, and RQDs-dAb-CA724. Finally, as shown in Scheme 2, Figures S4 and 5a, cAb-MBs were successively incubated with tumor markers and then GQD-dAb or RQD-dAb, hence forming beads-based sandwich immune

structure DSBs (GQDs-CEA-SiO$_2$-L, RQDs-CA125-SiO$_2$-L, GQDs-SCCA-PS-L, RQDs-AFP-PS-L, GQDs-NSE-PS-S, RQDs-CA724-PS-S), which could be utilized for ultrasensitive multiplexed analysis of tumor biomarkers on flow cytometry.

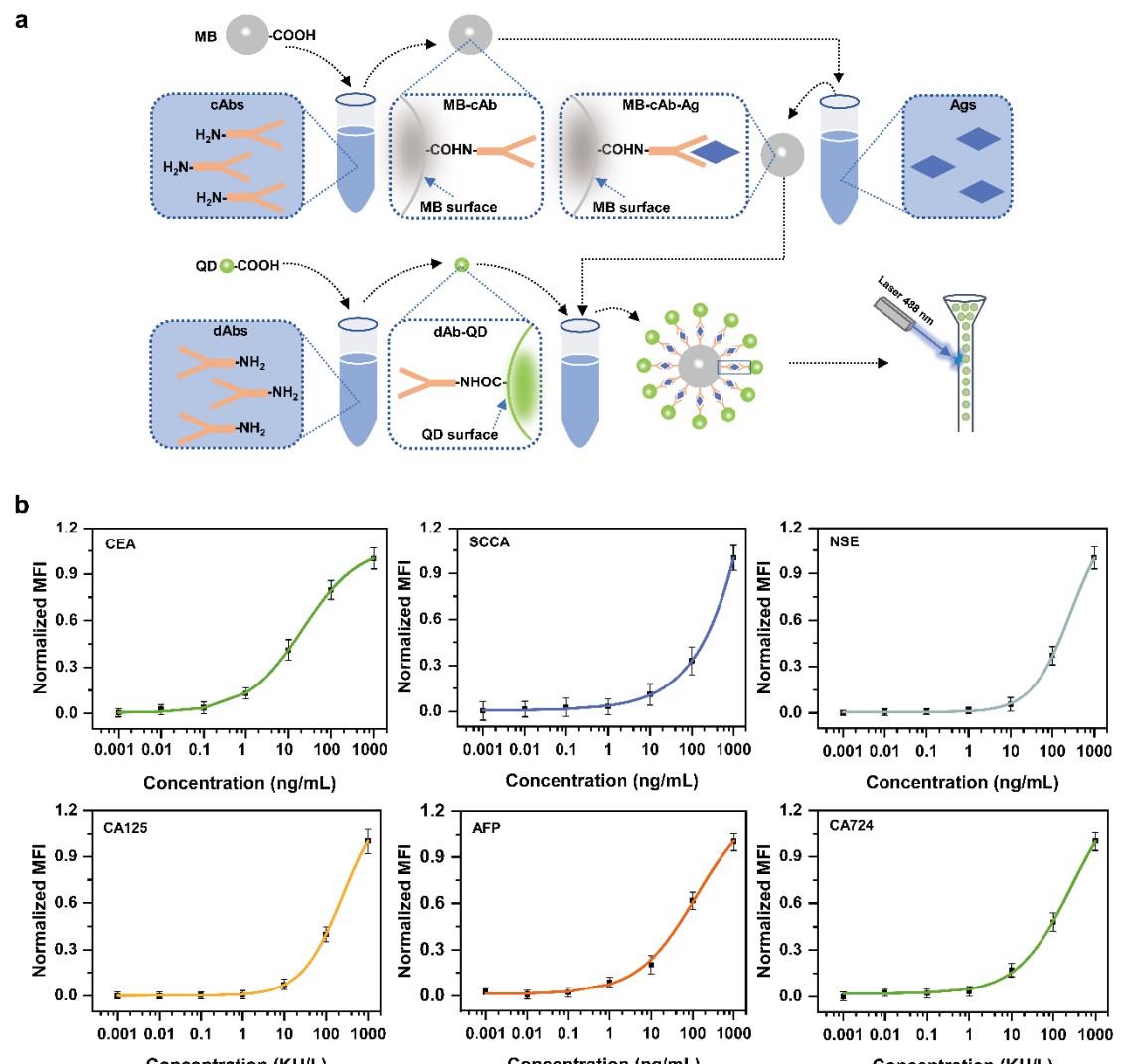

**Figure 5.** (**a**) Schematic illustration for the sandwich immunoassay of tumor markers. (**b**) Standard curves of CEA, SCCA, NSE, CA125, AFP, and CA724 in 6-plexed tumor markers assay.

In a single bottle test, samples containing no tumor marker (blank), each of the six tumor markers, or a mixture of the six tumor markers, were accurately evaluated by conducting a parallel assay on flow cytometry. It is worth noting that all the barcodes were simultaneously excited by the blue 488 nm laser, then the encoded signals and the detection signals were obtained from FSC, SSC, FL1-H (GQDs), and FL2-H (RQDs) channels at the same time, which shows that this platform exhibited simple and economical detection capacity.

From Figure 5b, it can be seen that the results of the six model targets of tumor markers which are the detection signal values of each target in the FL1-H or FL2-H channel progressively increase as the concentration of tumor markers increased in a large range, which are attributed to the selective binding between the corresponding antibody and antigen in multiplex sandwich immunoassays. Moreover, the four-parameter fitting standard curve of the six-plex tumor markers tests after optimization are shown in Figure 5b, which shows a good linear relationship between analyte concentration and the detection signal

value (normalized MFI), as well as excellent quantitative analysis performance of barcodes platform to tumor markers. Detection limits of 0.05 ng/mL for CEA, 0.92 KU/L for CA125, 0.28 ng/mL for SCCA, 0.09 ng/mL for AFP, 0.16 ng/mL for NSE, and 0.36 KU/L for CA724 are achieved. Three replicates were used for each experiment. The respective limits of detection (LOD) of six tumor markers exhibited ultrahigh sensitivity, which is superior to the other reported methods (Table S1, supplementary materials) [18,36]. Hence, the microbeads-based DSBs platform can be applied as excellent detection performance tool for highly-sensitive multiplexed analysis of tumor biomarkers, providing a novel way to reduce signal interference between barcode and reporter in SAT.

## 4. Conclusions

In summary, we report an ingenious method to eliminate the bleed-through effect of EFS in the DFS channel of flow cytometry by using low-background DSBs based on MBs and QDs. The low-background DSBs were prepared via combination strategy of two signals containing scatter signals and fluorescence signals. Three types of MBs ($SiO_2$-L, PS-L, and PS-S) were distinguished by the scattering channel of flow cytometer (FSC vs. SSC) to obtain the scattered signals. GQDs or RQDs were coupled to the surface of MBs by sandwich immune structure to obtain the distinguishable fluorescent signals. We therefore applied a combination strategy of the two signals to generate six distinguishable sandwich-structured low-background DSBs, containing GQDs-CEA-$SiO_2$-L, RQDs-CA125-$SiO_2$-L, GQDs-SCCA-PS-L, RQDs-AFP-PS-L, GQDs-NSE-PS-S, and RQDs-CA724-PS-S. Furthermore, the amount of conjugated capture antibody on the MBs surface was optimized by comparing the change of detection sensitivity with the addition of capture antibody for effective establishment of the highly sensitive suspension assay. The optimum addition amounts of cAb-CEA, cAb-CA125, cAb-SCCA, cAb-AFP, cAb-NSE, and cAb-CA724 are 500, 1440, 1320, 640, 500, and 1000 ng, respectively. The combination measurements of specificity and NSB in SAT platform were carried out by incubating the cAb-MBs with individual QDs-dAbs. Finally, the high-sensitivity analysis abilities of the DSBs to tumor markers were demonstrated via multiplex sandwich immunoassays. The detection limits of CEA, CA125, SCCA, AFP, NSE, and CA724 are 0.05 ng/mL, 0.92 KU/L, 0.28 ng/mL, 0.09 ng/mL, 0.16 ng/mL, and 0.36 KU/L, respectively. SAT platforms were established for simultaneous multiplexed detection of CEA, CA125, SCCA, AFP, NSE, and CA724. The SAT platform can be applied as an excellent detection performance tool for highly-sensitive multiplexed analysis of tumor biomarkers. These developed DSBs, with low background signal for high-sensitivity analysis of multiple tumor markers, would offer a novel way to reduce signal interference between barcode and reporter in SAT for early, efficient and accurate diagnosis of cancer.

**Supplementary Materials:** The following are available online at https://www.mdpi.com/article/10.3390/chemosensors10040142/s1, Figure S1: Representative SEM images and microparticle size statistics of MBs. Figure S2: SEM images with high magnification of PS and $SiO_2$. Figure S3: The relationship between the PL peak wavelength of 522 nm (blue) and 630 nm (red) CdTe. Figure S4: Schematic representation of GQDs or RQDs were coupled to the surface of MBs ($SiO_2$-L, PS-L and PS-S) by sandwich immune structure. Figure S5: Optimization of added amounts of cAb. Figure S6: Schematic illustration of combinatorial experiments of 6-plex specific and non-specific binding. Figure S7: Combinatorial experiments of 6-plex specific and non-specific binding. Table S1: Limit of detection (LOD) of six-plexed tumor markers biodetection compared to other the data of SAT.

**Author Contributions:** Conceptualization, B.Z. and S.-N.D.; methodology, B.Z. and S.-N.D.; validation, B.Z. and W.-S.T.; investigation, B.Z.; resources, S.-N.D.; data curation, B.Z. and W.-S.T.; writing—original draft, B.Z.; writing—review and editing, B.Z. and S.-N.D.; supervision, S.-N.D.; project administration, S.-N.D.; funding acquisition, S.-N.D. All authors have read and agreed to the published version of the manuscript.

**Funding:** This work is supported by the National Key Research and Development Program of China (2017YFA0700404), the National Natural Science Foundation of China (22174015).

**Institutional Review Board Statement:** Not applicable.

**Informed Consent Statement:** Not applicable.

**Data Availability Statement:** Not applicable.

**Acknowledgments:** The authors thank the instrument management staff of the School of Chemistry and Chemical Engineering, Southeast University for their support.

**Conflicts of Interest:** The authors declare no conflict of interest.

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
