# Peer review of "Dual-Signal-Encoded Barcodes with Low Background Signal for High-Sensitivity Analysis of Multiple Tumor Markers"

_chemosensors, doi:10.3390/chemosensors10040142_

Round 1

Reviewer 1 Report

In this article, the authors designed dual-signal sandwich immunoassay based on suspension assay technology to detect multiple tumor markers. In this approach, the bleed-through effect was eliminated and background noise was reduced. The topic is interesting whereas the manuscript should be further improved before acceptance of publication. Below are several questions that might be helpful to authors.  

  1. Full names of GQD and RQD should be given in abstract and manuscript. Also, grammars throughout the manuscript should be checked and corrected. For example, in 4. Conclusion: to effective establishment of ...
  2. Scale bars should be indicated in laser scanning microscope images.
  3. Detection limit of this approach is ng/ml scale. What is the superiority of this approach compared to elisa kit which easily gives pg/mL sensitivity?
  4. I wonder how stable the sandwich structure could be in complicated biological/pathological samples? Will it be influenced by salt/ionic strength/serum/cell debris?

Author Response

Responses to the reviewers' comments

The authors appreciate the reviewers and the editors for their valuable comments. All of them were carefully considered in the attached revised version. Comments are written in blue and responses of the authors in black.

Comments from the reviewers

Reviewer 1

In this article, the authors designed dual-signal sandwich immunoassay based on suspension assay technology to detect multiple tumor markers. In this approach, the bleed-through effect was eliminated and background noise was reduced. The topic is interesting whereas the manuscript should be further improved before acceptance of publication. Below are several questions that might be helpful to authors. 

  1. Full names of GQD and RQD should be given in abstract and manuscript. Also, grammars throughout the manuscript should be checked and corrected. For example, in 4. Conclusion: to effective establishment of ...

Response: We appreciate the reviewer for the valuable suggestion. According to your suggestion, the full names of GQD and RQD have been indicated in the abstract and manuscript. Besides, grammars throughout the manuscript have been checked and corrected.

  1. Scale bars should be indicated in laser scanning microscope images.

Response: Thanks a lot for your advice. Based on your suggestion, we have added scale bars to all laser scanning microscope images. (Please see Scheme 2, Figure 2h and Figure 3h in the revised version).

  1. Detection limit of this approach is ng/ml scale. What is the superiority of this approach compared to elisa kit which easily gives pg/mL sensitivity?

Response: The superiority of this approach compared to elisa kit is applicable to the high-throughput, simultaneous detection of multiple analytes within a small, single sample volume.

  1. I wonder how stable the sandwich structure could be in complicated biological/pathological samples? Will it be influenced by salt/ionic strength/serum/cell debris?

Response: According to the previous serum experiments of our group, the sandwich structure like this has good stability.

Reviewer 2 Report

I think this is an interesting paper with well designed experiments and interesting results.  I do think the paper could be improved to make it clearer to the reader what the impact of this work is, and to make it easier to understand and follow.

  • Overall needs a spelling and grammar check, it is a little difficult to read through at times.
  • Some paragraphs are much too long, and should be broken up into several smaller paragraphs to improve readability.
  • Introduction should be stated in future tense of what will be done framed as a hypothesis, as opposed to what was done.
  • What exactly makes this method unique? Has a sandwich type assay like this not been done before?  Is it the specific interactions of MBs and QDs?  It’s not entirely clear from the introduction and discussion what part of the method is new and unique.
  • Scheme 1 needs a picture legend explaining what each part is (similar to scheme 2).
  • Some acronyms are not defined on first use, but need to be (such as PVP).
  • Figure 3H is difficult to see, brightness should be increased so the red is more visible.
  • Section 3.4 - Limits of detection should be directly stated, as well as the number of replicates used.
  • What was the solvent used in these experiments (water, PBS, serum)? So far as I saw you just said solution, which is not clear enough what exactly was used.
  • Has this test been verified in biological solution, such as urine or serum, which is what it would be performed in in clinical analysis.
  • Conclusions - What is the future work and next steps that stem from this work?

Author Response

Responses to the reviewers' comments

The authors appreciate the reviewers and the editors for their valuable comments. All of them were carefully considered in the attached revised version. Comments are written in blue and responses of the authors in black.

Comments from the reviewers

Reviewer 2

I think this is an interesting paper with well designed experiments and interesting results. I do think the paper could be improved to make it clearer to the reader what the impact of this work is, and to make it easier to understand and follow.

Overall needs a spelling and grammar check, it is a little difficult to read through at times.

Response: We appreciate the reviewer for the valuable suggestion. According to your suggestion, spelling and grammar have been checked and corrected.

Some paragraphs are much too long, and should be broken up into several smaller paragraphs to improve readability.

Response: According to your question, some paragraphs that were too long have been broken up several smaller paragraphs.

Introduction should be stated in future tense of what will be done framed as a hypothesis, as opposed to what was done.

Response: According to your suggestion, The last sentence of the introduction have been stated in future tense.

What exactly makes this method unique?

Response: The signal combination method of scattering signal and fluorescence signal not only avoids the bleed-through effect of the encoding fluorescence signals (EFS) and detection fluorescence signals (DFS) in the corresponding flow cytometer channel, but also realizes the simultaneous detection of multiple targets.

Has a sandwich type assay like this not been done before? 

Response: Sandwich assays based on this signal combination method generated fluorescent signals that have the dual function of identifying and quantifying analyte targets.

Is it the specific interactions of MBs and QDs?  It’s not entirely clear from the introduction and discussion what part of the method is new and unique.

Response: The structural parameters (e.g., size and component) of MBs (SiO2-L, PS-L and PS-S) into scattering signals in flow cytometry is new and unique.

Scheme 1 needs a picture legend explaining what each part is (similar to scheme 2).

Response: Thanks for the reviewer's helpful advice. We have added a picture legend in Scheme 1.

Some acronyms are not defined on first use, but need to be (such as PVP).

Response: Thanks a lot for your advice. Based on your suggestion, all acronyms were defined on first use.

Figure 3H is difficult to see, brightness should be increased so the red is more visible.

Response: For more visibility, brightness of Figure 3H have been increased.

Section 3.4 - Limits of detection should be directly stated, as well as the number of replicates used.

Response: Thanks a lot for the valuable suggestion. Limits of detection and the number of replicates used have been directly stated in Section 3.4.

What was the solvent used in these experiments (water, PBS, serum)? So far as I saw you just said solution, which is not clear enough what exactly was used.

Response: The solvent used in these experiments was PBS.

Has this test been verified in biological solution, such as urine or serum, which is what it would be performed in in clinical analysis.

Response: This test has not been validated in biological solutions.

Conclusions - What is the future work and next steps that stem from this work?

Response: This test will be validated in serum.